# Comparing the fit of N95, KN95, surgical, and cloth face masks and assessing the accuracy of fit checking

**Eugenia O'Kelly**[1]*, **Anmol Arora**[1], **Sophia Pirog**[2], **James Ward**[1], **P. John Clarkson**[1]

**1** Cambridge University, San Francisco, CA, United States of America, **2** Northwestern University, Evanston, IL, United States of America

* eo339@cam.ac.uk

## Abstract

### Introduction

The COVID-19 pandemic has made well-fitting face masks a critical piece of protective equipment for healthcare workers and civilians. While the importance of wearing face masks has been acknowledged, there remains a lack of understanding about the role of good fit in rendering protective equipment useful. In addition, supply chain constraints have caused some organizations to abandon traditional quantitative or/and qualitative fit testing, and instead, have implemented subjective fit checking. Our study seeks to quantitatively evaluate the level of fit offered by various types of masks, and most importantly, assess the accuracy of implementing fit checks by comparing fit check results to quantitative fit testing results.

### Methods

Seven participants first evaluated N95 and KN95 respirators by performing a fit check. Participants then underwent quantitative fit testing wearing five N95 respirators, a KN95 respirator, a surgical mask, and fabric masks.

### Results

N95 respirators offered higher degrees of protection than the other categories of masks tested; however, it should be noted that most N95 respirators failed to fit the participants adequately. Fit check responses had poor correlation with quantitative fit factor scores. KN95, surgical, and fabric masks achieved low fit factor scores, with little protective difference recorded between respiratory protection options. In addition, small facial differences were observed to have a significant impact on quantitative fit.

### Conclusion

Fit is critical to the level of protection offered by respirators. For an N95 respirator to provide the promised protection, it must fit the participant. Performing a fit check via NHS self-assessment guidelines was an unreliable way of determining fit.

**Data Availability Statement:** Raw data and analysis of the N95 qualitative and fit check tests are available on the Cambridge University open access repository Apollo (DOI: 10.17863/CAM. 56361).

**Funding:** The author(s) received no specific funding for this work.

**Competing interests:** The authors have declared that no competing interests exist.

# Introduction

During the course of the COVID-19 pandemic, the importance of face mask fit has become apparent, while testing to ensure fit has decreased [1, 2]. It is known that respiratory protective equipment is only effective when there is an adequate seal formed between a mask and the person's face to ensure that inhaled air is actually filtered. Indeed, research has suggested that an ineffective seal is the primary cause of airborne contamination amongst wearers of face mask [3]. It has been noted that leakage around the face mask may account for a third of airflow across surgical masks and a sixth of the flow across respirators [3]. Fit is recognized as being particularly important when determining whether masks are capable of reducing the spread of fine particles. It is normal practice for respirators to be fit tested before use in clinical practice.

COVID-19 has strained supply chains of masks and fit testing supplies while simultaneously placing heavy workloads on hospital staff. This has led to many healthcare facilities having to abandon normal fit testing procedures [1]. Qualitative and quantitative fit testing has been replaced by self-performed fit checks, in which the user feels for air leaks [4]. The potential impact of abandoning or replacing fit testing procedures on respiratory protection with the current self-assessment remains understudied. Prior research has evaluated similar self-assessment procedures and found them ineffective [5], further raising concerns over the safety of the new self-assessment procedures.

For individuals outside of healthcare facilities, the Center for Disease Control and Prevention (CDC) in the United States and Public Health England (PHE) in the United Kingdom have advised the general public to wear fabric face coverings in public [6, 7]. Notably, there is little literature exploring the fit of fabric face coverings and how much protection they may offer wearers where fit issues are present. Even in normal times, fit testing is reserved for the use of N95 respirators and research into the importance of fit for other mask types, such as surgical masks, is limited.

Our study explores the fit and associated protection offered by a range of face masks types. Notably, we test a pre-COVID-19 manufactured N95 respirator; a KN95 respirator, a standard surgical mask, and a selection of fabric face coverings. We aim to elucidate importance of fit for protecting the wearer, how well a simple fit check predicts fit, and the relative degree of protection offered by each mask type.

# Methods

## Quantitative fit testing

Two industry standard methods exist to evaluate the fit of masks: qualitative fit testing and quantitative fit testing. Qualitative fit testing is a subjective method in which the subject reports their ability to taste or smell a solution while wearing a mask. While qualitative fit testing is a NIOSH (National Institute for Occupational Safety and Health) approved method of testing the fit of N95 respirators, it has been previously shown to have a high false-positive rate [8]. Quantitative fit testing is an objective method of assessing fit and is the preferred standard when exact measurements are needed. For this study, quantitative fit testing was used to determine actual mask fit. Quantitative fit testing continuously measures the concentration of particles inside and outside a mask while it is worn (see Fig 1). For a mask with an established level of filtration ability, such as an N95 or KN95 respirator, a higher number of particles inside of the mask is indicative of poor fit. When gaps are present in the fit of the mask, unfiltered air is allowed to enter the mask, raising particle levels. Quantitative fit testing machines use these particle concentrations to calculate a fit factor via a standard formula [9].

Fit factor scores provide a quantitative measure of the degree of protection wearers might expect. While fit factor scores generated in a testing environment do not exactly predict the degree

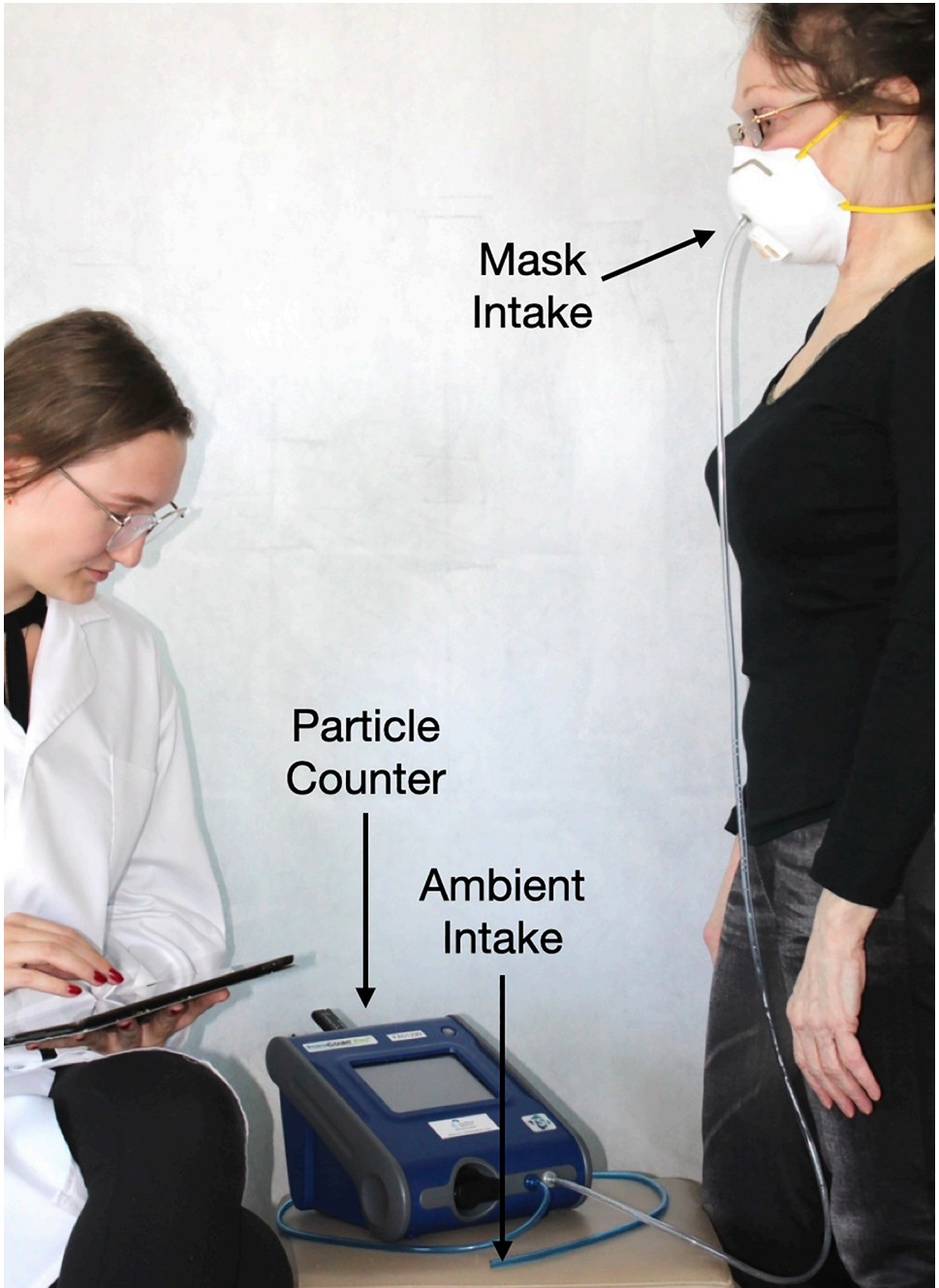

**Fig 1. Experimental setup for quantitative fit testing.** Particle counter measures particle concentrations in the mask from mask intake valve (clear) and from the ambient air by ambient intake (blue) while mask is worn by participant. Research assistant records readings and instructs participants in testing procedures. Not shown in this image: TSI model 8026 particle generator.

of workplace protection, fit factors have been shown to be an effective method of predicting actual workplace protection [10]. Higher fit factor scores indicate greater levels of protection. According to OSHA guidelines, half face respirators, such as N95 respirators, must achieve a fit of at least 100 to be worn as protective gear, according to OSHA guidelines [11]. The maximum score achievable when testing masks with less than 99% efficiency is over 200, represented as 200+.

Quantitative fit testing was performed with a TSI PortaCount Pro Respirator Fit Tester model 8038+, capable of assessing masks with less than 99% filtration efficiency. The Portacount 8038 measures particles with a minimum size of 0.02 micrometers at a sampling flow rate of 350 cm^3/min. Fit tests were conducted using OSHA protocol 29CFR1910.134 [12]. A minimum fit factor of 100 must be achieved for N95 respirators to offer appropriate protection according to OSHA guidelines [11]. A fit factor of 200+ represents the highest score possible for the masks tested. Quantitative fit testing with the PortaCount 8038+ has a fit factor error range of +/- 10% of reading.

While quantitative fit testing is a highly accurate method of assessing fit, the procedure needed to prepare a mask for testing renders it unusable for protective purposes after the test is completed. To take air samples from inside the mask, each mask to be tested must be fit with a testing gasket. This gasket places a permanent hole in the mask. To conserve N95 respirators during this international shortage, the number of participants tested in the study was limited, to match the number of available N95 respirators.

## Masks tested

Three categories of face masks were tested for fit: five N95 respirators from different manufacturers, a KN95 respirator, surgical style masks, and a selection of fabric face coverings. The N95 respirators tested are listed in Table 1. Each participant completed one 7-activity quantitative fit test and one subjective NHS procedural fit check per respirator.

The KN95 respirator was a Zhong Jian Le KN95 respirator, manufactured by Chengde Technology Co LTD, China and certified according to Chinese standard GB2626-2006 (seen in Fig 2 as F). All KN95 respirators follow a very similar design, with two panels joining at the center of the face and a short nose wire over the bridge of the nose.

Five basic fabric masks were tested on three of the participants. These included a 100% cotton bandana, a mask made of stretch material, a pleated surgical-style mask, and two masks designed to contour to the face. All of these masks were of simple construction, made of one or two layers of fabric and containing neither a filter nor fitting aids such as a nose wire.

N95 and KN95 respirators were worn for at least five minutes before testing to purge interior particles, as recommended in the quantitative fit testing guidelines [12]. Surgical masks and fabric face coverings, which are considered non-sealing and thus do not apply to such recommendations, were worn for at least three minutes before testing.

Five basic fabric masks were tested: a cotton bandana, a double-layered surgical-style pleated mask, a mask made of stretch material, and two masks designed to contour to the face. All masks were of basic construction and did not include insert filters.

## Participants

Seven participants took part in this study. This study was approved by the Cambridge University Engineering Department Ethics Review Committee and written consent obtained from all participants. The individual in this manuscript has given written informed consent (as

**Table 1. Model number, manufacturer, manufacturer information, country of manufacture, and NIOSH approval numbers for the five N95 respirators tested.**

| Model | Manufacturer | Manufacturer Headquarters | Country of Manufacture | NIOSH Approval (TC) Number |
|---|---|---|---|---|
| 8511 | 3M | Saint Paul, USA | USA | TC-84A-1299 |
| 8200 | 3M | Saint Paul, USA | USA | TC-84A-4271 |
| AP0028 | Aero Pro Co Ltd | Taipei, Taiwan | Taiwan | TC-84A-4175 |
| 9500 | Makrite | Tianzhong, Taiwan | China | TC-84A-5411 |
| ZYB-11 | Xiantao Zong | Wuhan, China | China | TC-84A-7877 |

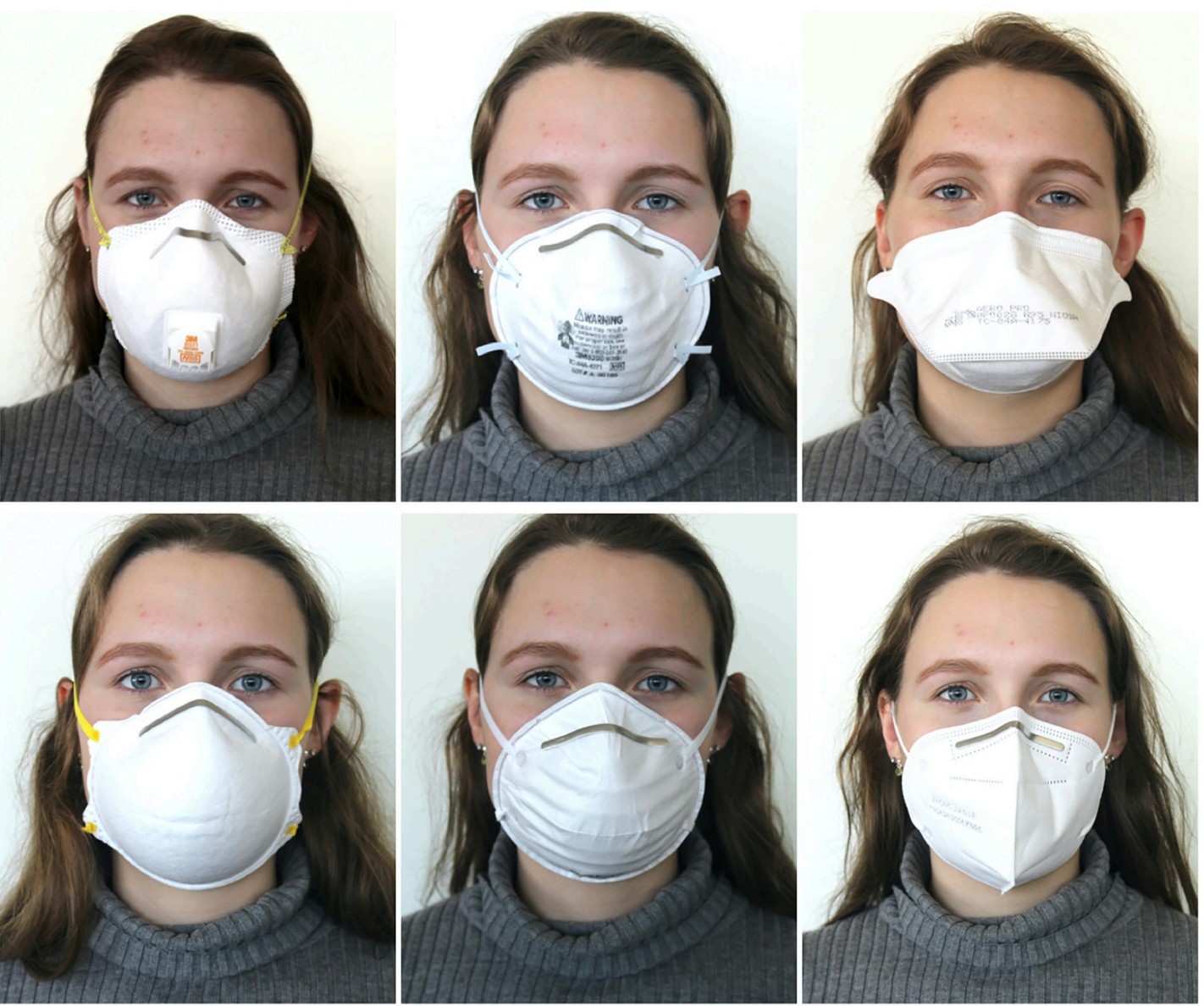

**Fig 2. N95 and KN95 respirators tested.** Top row, from left to right: 3M 8511, 3M 8200, Aero Pro AP0028. Bottom row, from left to right: Makrite 9500, Xiantao Zong ZYB-11, Zhong Jian Le KN95.

outlined in PLOS consent form) to publish these case details. To minimize infection risk for both participants and researchers, participants were recruited from two groups. Within each group, members had regular contact with each other. Three participants were male and four participants were female. Participant ages ranged from 17 to 74, with a mean age of 45 and a median age of 51. Participants completed seven activities as defined by the OSHA fit-testing protocol. This range of activities is intended to represent the spectrum of normal occupational activity, and thus reproduce fit during daily activity.

The use of groups closely related also allowed us to compare the impact of minor changes in facial feature by comparing the fit of mother and daughter in two cases. Participant F-68 was the mother of Participant F-28 and the two were remarked to have very similar

facial structure. Participant F-51 was the mother of Participant F-18, with similar facial structure.

Two modifications were made to the 29CFR1910.134 protocol. First, participants were allowed to sit through sections of the fit testing. Secondly, participants were allowed to adjust their masks if needed, with the researcher making note if adjustments were necessary during the fit test. This allowance was made to allow us to complete testing on poorly fitting masks. Information on whether adjustments were made during testing can be found in the openly available data set.

Two Participants, M-51 and M-29, had some degree of facial hair or stubble. Participant M-51 has the presence of a short beard and Participant M-29 had moderate stubble. These participants were included to (1) assess if the presence of facial hair affects fit checks and (2) assess if facial hair affects the fit of KN95, surgical, and fabric face masks as well as N95 respirators.

### Fit checks

Participants were asked to don the mask and perform a fit check according to UK National Health Service (NHS) guidelines [13]. Participants were asked to feel around the sides of the mask and report if they could feel any air leaks, suggesting poor fit. They were also instructed to feel if the mask responded to heavy breathing, with the mask being pulled more tightly against the face during a sharp inhale. Participants reported the result of the fit check (either believing the mask fit properly, or a believing the mask did not fit properly). They were then asked how confident they were in their assessment, with "high" denoting a high level of confidence that their prediction of mask fit was correct, "medium" denoting a medium level of confidence, and "low" denoting little confidence in their fit check result. To assess the reliability of the fit-checking method, fit check answers were compared with the quantitative fit factor for each mask.

## Results

### Protection and fit

The 3M model 8511 N95 respirators fit the greatest number of participants (see Fig 3). Three out of the seven participants achieved a 'pass' value while wearing the 3M 8511 respirators, two of whom received the maximum fit factor. Four of the participants failed the fit test with the 3M 8511 model.

The Xiantao Zong respirator and Aero Pro respirator fit none of the participants. The Xiantao Zong respirator had fit factors with a mean of 13.2 and a median of 5.6. The Aero Pro had a mean of 35.5 and a median of 21. The 3M 8200 fit two out of the seven participants, with a mean of 72.3 and a median of 64. The Makrite fit one out of the seven participants with a mean of 37.7, a median of 14.

The KN95 respirator had a visible poor fit and showed very low scores across all participants, with an average fit factor of 2.2. Minimal variation was experienced between participants.

Surgical masks are not designed to seal to the face and thus do not provide wearers with the same level of protection as an N95 respirator. Nonetheless, our results indicated that surgical masks blocked over 40% of particles for all participants in the last cycle of the quantitative fit test (see Fig 3). The surgical mask showed similar fit factor to the KN95 respirator with an average fit factor of 3.2 and median fit factor of 2.2. This average fit factor was within the lower range observed by Oberg et all in his 2008 study [14]. This lower score may be due to the grade or quality of the masks used. Although marketed and sold as medical grade, there was some question as to if the surgical masks, manufactured in China, actually met necessary standards.

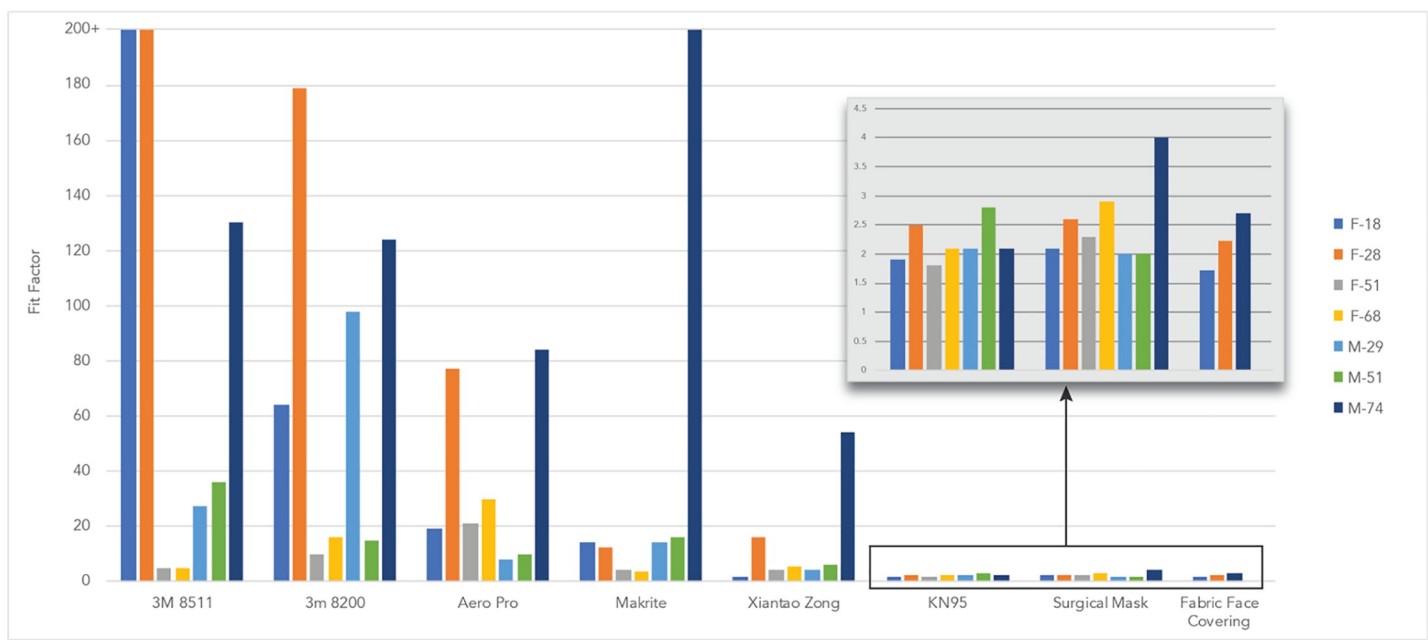

**Fig 3. The fit factor achieved by a set of volunteers while wearing different varieties of masks.** Protection when wearing an N95 respirator was high only if the respirator properly fit the participant. Fit factors for KN95 respirator, surgical masks, and cloth masks were similar.

Inside the box of masks, a certificate was included which denied the masks were medical grade. Conversely, the exterior package indicated the masks were of medical-grade. Discussion with the chief of staff of a local hospital revealed this to be a common problem in imported masks during COVID-19.

The fabric face masks tested produced similar fit factors to commercial KN95 respirator and surgical masks; however, the fabric masks used were of the most basic design and construction. It is likely that an improved design, coupled with the use of better materials, would increase the effectiveness of these masks. The fabric face masks had an average fit factor of 2.1 and a median factor of 2.1. Fit factor was similar for all masks and all participants, with a spread of 1.5.

While fit factor scores are derived from the number of particles filtered by the mask, the fit factor numbers are not directly equal to filtration efficiency (percentage of particles filtered through the mask material). Measurements of particles counts inside and outside of the mask were compared and are illustrated in Fig 4, which represents the number of interior vs. exterior particles in the last cycle of the test.

## Impact of minor facial differences

Minor differences in facial features were seen to have a significant impact on fit. Participant F-68 was the mother of Participant F-28. Their facial structures appeared very similar, with similar bone structure and nose length and width. F-28 had a maximum width of 113 mm and F-68 having a maximum width of 110 mm where the respirator contacted the cheek. Nose measurements were almost identical, with lengths of 30 (F-28) mm and 30 mm (F-68) mm and alar widths of 29mm and 30mm respectively.

However, the fit factor scores achieved by the two individuals bore no correlation. This surprised us to such a degree that we repeated fit tests for the respirators which fit F-28 but not F-68 five times on each participant. Repeated tests continued to show the same inconsistencies

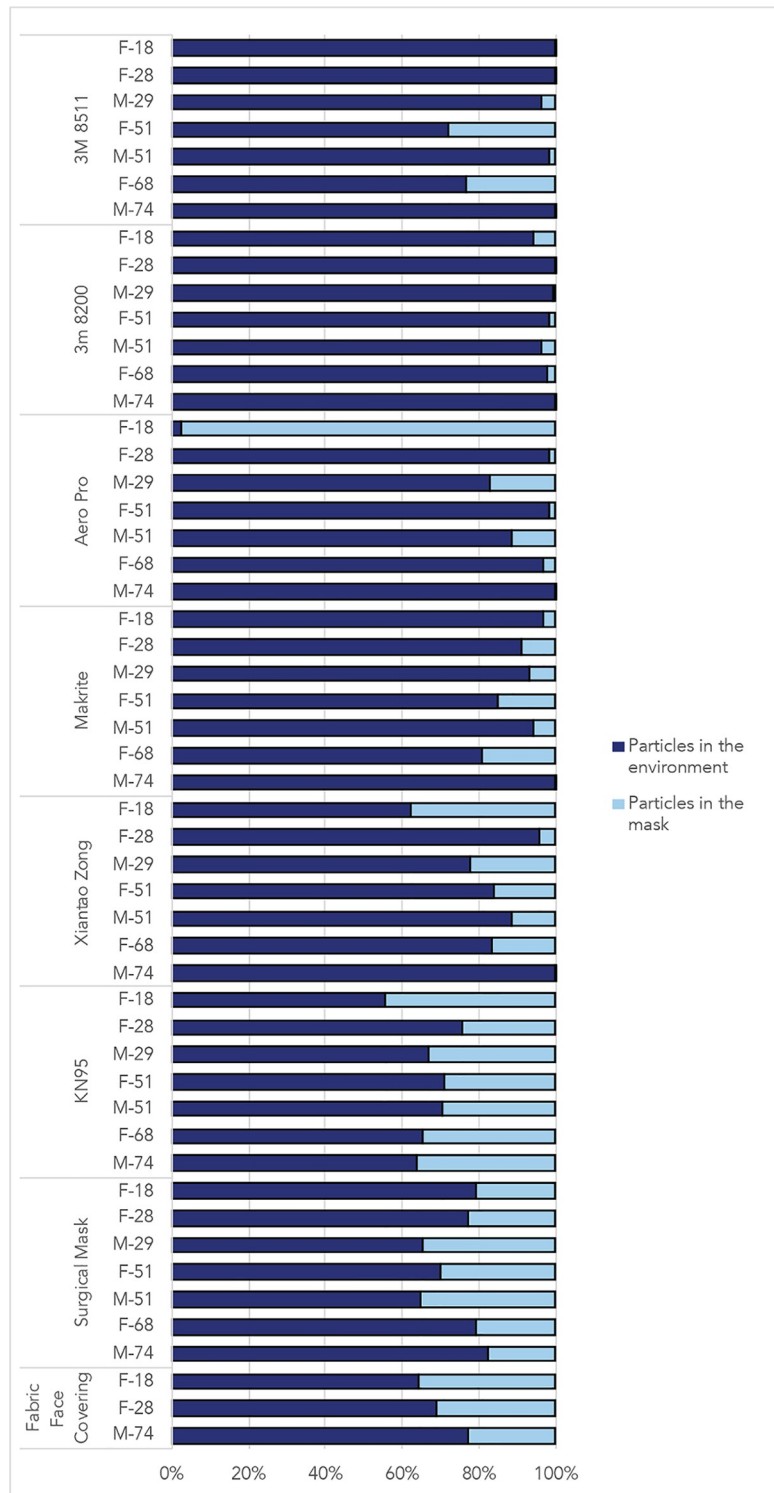

**Fig 4. Percentage of particles outside of the mask and percentage of particles measured inside of the mask during the last cycle of normal breathing during the fit test.**

in fit, with F-28 scoring 200+ on every test and the quantitative fit scores for F-68 falling close to +/- 10% of the initial fit test. While 8511 and 8200 achieved maximum fit scores on F-28, they achieved only minimal fit scores on F-68. Participant F-28 scored over 200 on the 3M 8511 respirator and 178 on the 3M 8200 respirator, while participant F-68 scored only 4.8 on the 3M 8511 and 16 on the 8200. A frontal visual inspection failed to reveal the source of the difference. However, an in-depth external inspection of a respirator fit during a series of activities revealed the chin of the wearers to be the major differentiating factor (see Fig 5). Participant F-28, being younger, retained more subcutaneous fat in the skin under the chin. This extra padding allowed the two respirators to seal when the wearers jaw was tightly shut or when swallowing. While impactful, caliper measurements showed the subcutaneous fat to only differ by approximately 3mm–with 8.3 mm average of subcutaneous fat in F-28 and 5.1 average in F-68. With this slight difference in under-chin padding, Participant F-68 had intermittent gaps during certain activities.

Fit differences between Participant F-51 and Participant F-18 showed a similar pattern. Participant F-18's youth led to higher amounts of subcutaneous fat which seemed to aid in the respirator sealing around the chin and cheeks. While we were not able to visually identify the fit issues experienced by F-51 with the same confidence, we believe the differences in subcutaneous fat or nose width (a matter of 3–4 mm difference) may have resulted in significant quantitative fit differences.

## Fit checks

All participants were able to correctly identify the lack of fit offered by the KN95 respirator, likely due to the visibly poor fit by which the respirator failed to sit flush to the face. Participants also correctly identified the lack of fit in surgical masks and fabric masks.

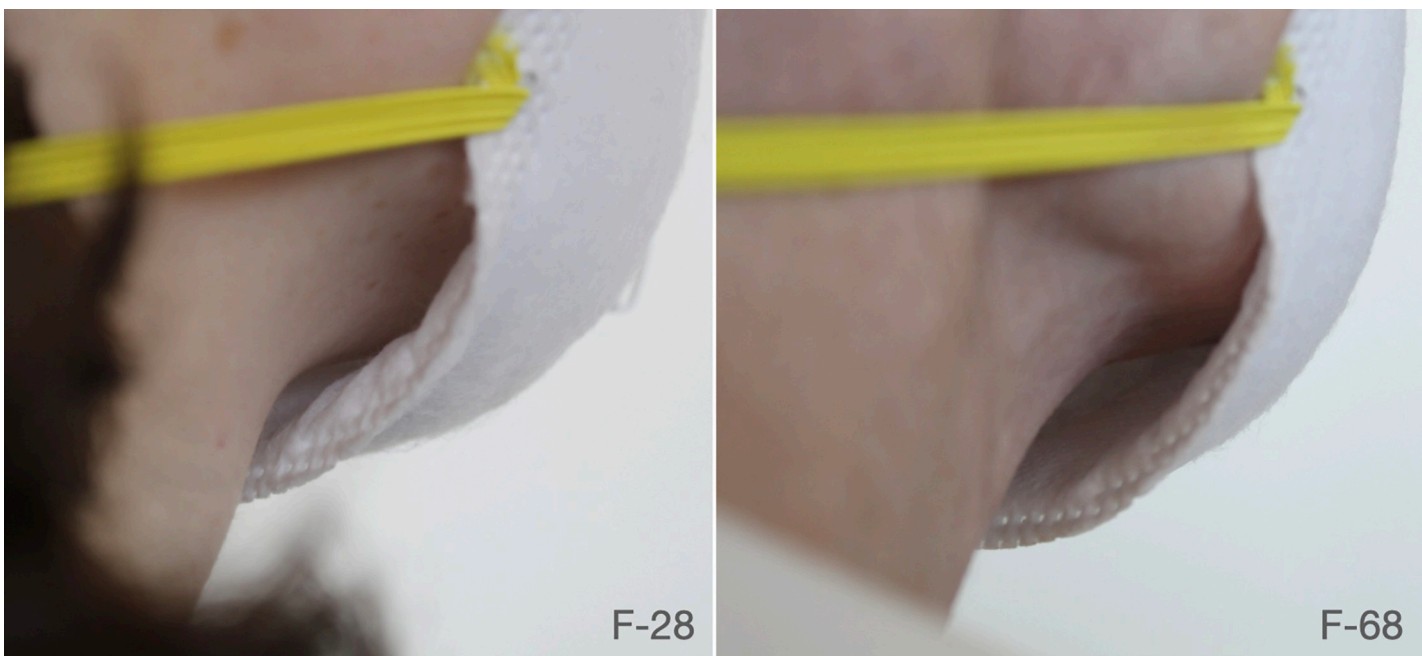

**Fig 5. The 3M 8511 mask on participants F-28 and F-68.** In both cases, the material of the respirator makes contact with the wearers' skin close to the tip of the chin, leaving what appears to be a visible gap. However, there was enough subcutaneous fat to consistently seal the mask in participant F-28, achieving the maximum score of 200+ for all tests. In the case of F-68, there was not enough subcutaneous fat to consistently ensure a seal between the respirator and the skin, leading to consistently low fit testing results.

Out of 35 tests on N95 respirators, participants believed 17 masks fit, 2 with low confidence, 7 with medium confidence, and 8 with high confidence. 6 of these respirators did indeed fit the wearer, leading to a 35% accuracy rate of predicting a respirator fit.

Out of the 35 tests, participants believed 18 respirators did not fit, 7 with medium confidence and 11 with high confidence. During all 18 of these tests, the testers themselves correctly identified that the respirator did not fit, thus leading to a 100% accuracy when predicting lack of fit.

If fit checks accurately predict fit, it is expected that the quantitative fit factor would be a function of the fit check results and fit check confidence. By extension, a respirator which passed the fit check would have a higher quantitative fit factor. In fact, this was not the case (Fig 6). The mean fit of respirators which were considered to fit with only a low degree of confidence was 138.5. The mean of respirators considered to fit with a medium degree of confidence was 74.1 while the mean of those believed to fit with a high confidence was 75.5. Even if only correct fit checks were taken into account, there was no correlation between the degree of confidence of fit and quantitative fit.

When considering respirators which did not fit, respirators which were believed to not fit with a high degree of confidence should have lower fit factors. Of the respirators which were correctly believed not to fit, those believed not to fit with a high degree of confidence had an average score of 18.2 while those believed not to fit with a medium confidence had a lower average of 13.4. Full breakdown of the data can be found in the related open data set.

## Discussion

### Summary of key findings

The importance of fit for N95 respirators has particular implications during the COVID-19 pandemic as these masks are generally reserved for clinicians at a time when hospitals are struggling to cope with demand for conventional fit testing. Proper fit is critical for N95 respirators to protect the wearer. Even a small fit issue not detected by the wearer when performing a fit check can greatly decrease the protection offered by the N95 respirator. Our results

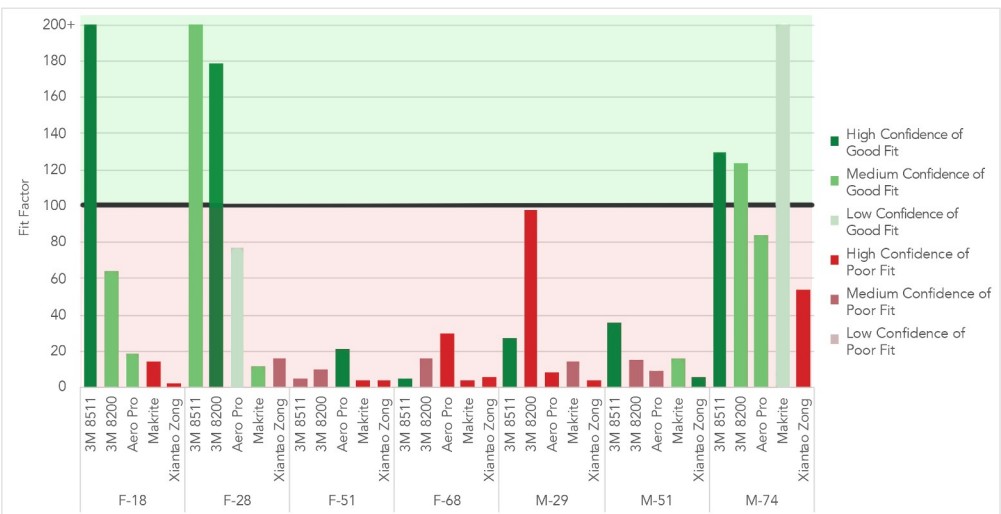

**Fig 6. Participants' fit-check predictions with the quantitative mask fit factor organized by participant.** Fit factor results are color coded to represent the participants' fit check results with green representing a belief of fit and red representing a belief the mask did not fit. Depth of color represented confidence with lighter shades representing low confidence in the fit check results and darker shades representing a high confidence.

indicate that the method of fit check, which is being used in many hospitals, is not reliable and has similar failure rates as previously tested self-assessment protocols [5]. The results show that 4 out of 7 participants were unable to achieve a proper fit from any of the tested N95 respirators. In addition, all participants made at least one incorrect determination of fit after performing a fit check. These findings suggest that traditional quantitative or quantitative fit testing for individuals in need of respiratory protection is highly advisable. The high proportion of fit test failures, and the associated reduction in effectiveness, has particular implications for healthcare facilities and other work-environments where fit testing may be abandoned during a pandemic.

These results indicate that proper fit is essential for N95 respirators to provide high degrees of protection. When fit was achieved, N95 respirators filtered more than 95% of airborne particles and offer superior protection. Poorly fit N95 respirators offered a range of protection, in some cases comparable with surgical and cloth masks.

This study confirms that NIOSH certification that a mask can perform at N95 levels alone is insufficient if the mask is poor fitting. Proper fit is absolutely necessary if the mask is to offer the wearer protection. Furthermore, our results indicate it is not enough to assume that any N95 respirator will be likely to fit the majority of a population. The most widely fitting mask, the 8511 N95, fit only three out of the seven participants. Other masks, such as the Aero Pro and Xiantao Zong, did not fit any of the participants adequately.

Another key finding was that surgical masks, KN95 respirators, and fabric masks offered similar levels of protection. There was minimal variation amongst participants in the fit offered between these three types of masks. Furthermore, facial hair and stubble seemed not to lead to the same reduction of fit in KN95, surgical, or fabric masks as is seen in N95 respirators.

## Implications of findings

In healthcare settings, clinicians are provided with N95 respirators or equivalent (e.g. FFP3) where procedures are being performed which are associated with a high risk of viral transmission. Existing literature is limited but has noted that poor fit can severely alter the effectiveness of clinical respiratory protective equipment, such as N95 respirators. Such research has indicated that better fit is associated with increased protection in controlled testing environments, but it also has been noted that this may not actually result in decreased rates of infection amongst clinicians who engage in fit testing [10]. In normal circumstances, the fit of a respirator is assessed by a professional fit test before clinical use in hospitals. Importantly, the pandemic has disrupted normal fit testing processes as there is both inadequate range and quantity of masks to satisfy conventional testing requirements [1]. There is also a limited range of alternative masks available in clinical settings in the event that an individual fails a fit test.

KN95 and surgical masks are also used in clinical settings, especially in some countries or when resources are limited such as during this COVID-19 pandemic. These masks are also relatively accessible and are increasingly used by the public. In the US, the Food and Drug Administration (FDA) and CDC have both approved the use of KN95 respirators where N95 respirators are unavailable, even though they are not routinely fit tested. Notably, concerns have been raised about the use of KN95 respirators, and in particular, the lack of an adequate seal [15]. Whilst the mask itself does have a high filtration efficiency, unless a tight seal is achieved, this may be rendered redundant. The KN95 respirator tested failed to fit any participant adequately enough to offer protective benefits above what might be achieved by a surgical or fabric mask.

It has been suggested that fit testing provides differential benefits for various types of masks, but general principles about face coverings are universal [16]. Respiratory protective

equipment can only perform at its stated material filtration levels when there is an adequate seal formed between a mask and the person's face. In addition, the space between the wearer's mouth and the mask acts as an extension of their breathing apparatus. During expiration, breathing movements increase pressure within the apparatus and force air through the filter of the mask. The reverse occurs during inspiration. If an ineffective seal is formed around the mask, contaminated air will take the path of least resistance through gaps around the mask, thus reducing the effectiveness of the mask substantially [17]. The findings of our research indicate that even if a mask is well constructed, you cannot predict the protection it will afford. Furthermore, anatomical differences which may appear insignificant, such as amount of subcutaneous fat under the chin, was found to have significant impact on fit. And some fit problems were only identified when the participant was engaged in a range of activities. All these elements of fit contribute to making the judgement of the fit of a mask particularly difficult and may have implications for the usability of 3D scanning technology for assessing mask fit.

For these reasons, it is critical to perform fit testing in order to ensure a respirator fits properly and is acting as an extension of the breathing apparatus rather than merely as a shield to block some of the incoming particles.

## The reliability of fit checks

Participants were unable to reliably predict whether respirators fit properly. While no participant mistook a well-fitting respirator for a poorly fitting respirator, participants routinely believed poorly fitting respirators fit well.

Nor did participants' confidence of a respirator's fit correlate with the actual fit of the respirator. Respirators believed to fit with a low confidence outperformed respirators which were believed to fit with a medium or high degree of confidence. Respirators which were believed with high confidence to have a poor fit performed on average better than those believed not to fit with only a medium degree of confidence.

These results suggest that individuals may not be able to accurately assess the fit of their mask through self-assessments, such as fit checks. Three out of the seven participants worked in a healthcare or healthcare-related field and had received a degree of mask fit education. One participant worked in a hazardous industry and had been required to wear face masks or respirators for certain working conditions. These participants were somewhat more likely to correctly identify mask fit through a fit check than the participants not familiar with mask wearing. However, all but one of these more experienced participants incorrectly identified 4 out of 5 N95 respirators. While our results indicate education and experience may be of benefit, a 20% fail rate in the experienced group is still concerningly high.

## Strengths and limitations

Several limitations affected this study. While quantitative testing is more accurate than qualitative testing, it destroys the respirators used. Due to limitations in respirator supply and need to preserve respirators for healthcare workers, we were only able to test a limited number of subjects. Another limitation came from participant recruitment. In order to protect participants and researchers, participants were recruited from protected groups. All participants recruited where Caucasian, although multiple ethnicities were captured including Polish, Irish, Scottish, Hungarian, German, and Swedish. Inclusion of other races would strengthen the study. Finally, due to the lengthy nature of quantitative fit testing and the age of some participants, most participants underwent the tests while seated; however, we do not expect this to have had an impact on the testing results.

Participants successfully represented a range of ages and prior experience wearing masks. Three participants had experience working in healthcare related fields, one participant had experience in an industry which can require the wearing of respirators and the remaining three participants did not have any relevant prior experience. Our results were generally concordant within our sample population; however, further studies with larger sample sizes would be necessary to offer definitive conclusions.

## Conclusion

To offer adequate respiratory protection for the wearer, a face mask must not only be made of high filtration, low resistance material, but must also fit the wearer. Fit was found to be difficult to visually or manually identify. Small anatomical variations were found to have major implications on fit. Some fit issues were only able to be identified when the participant engaged in a range of activities while an observer visually and manually inspected the mask.

This study also found that without proper fit, the value of high filtration material significantly decreases. Participants were seen to achieve similar protection from a fabric face covering as from a KN95 respirator, whose material should perform similarly to an N95 respirator. Indeed, our results indicated that surgical masks, poorly fit KN95 respirators, and basic fabric face coverings offered similar levels of protection to the wearer.

These findings support the argument that it is essential to ensure adequate fit of N95 respirators on healthcare workers, even during pandemics and times or shortage [5]. Having a wide variety of mask models and sizes stockpiled is critical as one mask model cannot be assumed to protect the majority of wearers. For civilian protection, no one type of mask proved superior to another. This is likely due to poor fit in the civilian mask options. Further research should focus on improving the fit of high-filtration options such as KN95 and surgical masks to capitalize on potential benefits offered by the materials such masks are constructed from.

## Supporting information

**S1 Text.**
(TXT)

## Acknowledgments

We would like to thank our participants who patiently sat through rounds of quantitative fit testing. Without their willingness to take time out of their busy lives and suffer through wearing uncomfortable masks, this study would not have been possible.

## Author Contributions

**Conceptualization:** Eugenia O'Kelly, James Ward, P. John Clarkson.

**Data curation:** Eugenia O'Kelly, Sophia Pirog.

**Formal analysis:** Eugenia O'Kelly, Sophia Pirog.

**Investigation:** Eugenia O'Kelly, Anmol Arora.

**Methodology:** Eugenia O'Kelly.

**Project administration:** Eugenia O'Kelly.

**Supervision:** James Ward, P. John Clarkson.

**Visualization:** Sophia Pirog.

**Writing – original draft:** Eugenia O'Kelly, Anmol Arora.

**Writing – review & editing:** Eugenia O'Kelly, Anmol Arora.

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
