## [Decision Letter · Decision Letter 0]

18 Sep 2020

PONE-D-20-25645

Comparing the Fit N95, KN95, Surgical, and Cloth Face Masks & Assessing the Accuracy of Fit Checking

PLOS ONE

Dear Dr. O'Kelly,

Thank you for submitting your manuscript to PLOS ONE. After careful consideration, we feel that it has merit but does not fully meet PLOS ONE’s publication criteria as it currently stands. Therefore, we invite you to submit a revised version of the manuscript that addresses the points raised during the review process.

We look forward to receiving your revised manuscript.

Kind regards,

Amitava Mukherjee, ME, Ph.D.

Academic Editor

PLOS ONE

Journal Requirements:

2. In your Methods section, please provide additional information about the participant recruitment method and the demographic details of your participants. Please ensure you have provided sufficient details to replicate the analyses such as:  a) a description of any inclusion/exclusion criteria that were applied to participant recruitment, b) a table of relevant demographic details, c) a statement as to whether your sample can be considered representative of a larger population, d) a description of how participants were recruited, and e) descriptions of where participants were recruited and where the research took place.

Reviewers' comments:

Reviewer's Responses to Questions

**Comments to the Author**

1. Is the manuscript technically sound, and do the data support the conclusions?

Reviewer #1: Yes

Reviewer #2: Yes

Reviewer #3: Partly

2. Has the statistical analysis been performed appropriately and rigorously? 

Reviewer #1: Yes

Reviewer #2: No

Reviewer #3: No

3. Have the authors made all data underlying the findings in their manuscript fully available?

Reviewer #1: No

Reviewer #2: Yes

Reviewer #3: No

4. Is the manuscript presented in an intelligible fashion and written in standard English?

Reviewer #1: Yes

Reviewer #2: Yes

Reviewer #3: Yes

5. Review Comments to the Author

Reviewer #1: This paper evaluates the level of fit offered by various types of masks, and most importantly, assess the accuracy of implementing fit checks by comparing fit check results to quantitative fit testing results. This work seems interesting, however there are some remarks and comments which improve the present quality of the paper.

Introduction: Please improve the state of art using more scientific papers. Honestly, the authors reviewed the recent articles in this subject but more attention to the scientific paper could improve the introduction parts.

Materials and methods: This section is not clear for readers. Please provide more comprehensive way to express the Fit method. For example, using a schematic. Moreover, why did you use this method? Why not other methods? Also, the structure of this section should be modified.

Results and discussion: Please improve the quality of the figures. In my opinion it is better to combine results and discussion sections. What do the authors think about the relation between diffusion phenomenon and Fit testing?

Reviewer #2: This manuscript introduced the fitting tests of N95, KN95, surgical and cloth face masks, and compared the fit check results to quantitative fit testing results, emphasizing the importance of fitting of face coverings. This study is related to the current situation of COVID-19, and provided important information on the proper usage and evaluation of respirators and face masks. However, a few important aspects regarding the testing of the fitting and data analysis still need further clarification. The reviewer, therefore, recommends a major revision of this manuscript.

General comments:

1. A schematic diagram of the experimental setup is needed to show how the measurement of fit score was conducted. Including photos of the masks will also be very helpful. It was mentioned that two KN95 respirators and a selection of fabric face coverings were tested, but only one set of results for each were provided. Moreover, which N95 mask is the “Xiantao Zong” mask? It was not listed in Table 1.

2. Proper sealing of the respirators requires that should be no facial hair between the mask and face. However, as mentioned in the “Strength and Limitations” section, “M-51 and M-29 had some degree of facial stubble or hair.” It is, therefore, not clear on the purpose of evaluating the fitting of respirators with these two participants.

3. The author introduced a fit check according to the UK National Health Service (NHS) guidelines and the details of choosing seven participants. Although it can cover all the range, as the author stressed that this study quantified the fitting test results, the number of samples (participants) is indeed not sufficient to reflect such efforts. Also, there are more subjective results due to limited participants, even if it represented a range of age and prior experience. It appears in Fig. 2 that M-74 and F-28 operated the respirators and face masks with higher filtration efficiencies. Are the deviations in mask filtration largely dependent on the skills of the participants in using the masks? Including more participants in the study would improve the results of this study.

4. It was mentioned that several N95 and the KN95 did not fit the participants because of loose fits. Was it because of the straps or the respirator itself? Were the straps adjustable? Agan, including photos of the masks and identifying the locations of leakage would be very helpful.

Specific comments:

1. “Fit factor” or “fit score”? Please consider unifying these terms.

2. Page 5, masks tested section, last paragraph: “N95 and KN95 masks were worn for at least five minutes before testing to purge interior particles. Surgical masks and fabric face coverings, which were non-sealing, were worn for at least three minutes before testing.” What is the reason for choosing 3 mins and 5 mins each? Can the author explain this or relevant references?

3. Page 6, quantitative fit testing section, second paragraph: it should be quantitative fit testing, not qualitative fit testing, in “Qualitative fit testing was performed with…”

4. Regarding the fit factor, how to measure mask particle concentration (Cr)? Please clarify this equation and the value of 100 and 200. In addition, please specify the experiment conditions of fit tests.

5. Why are the fit check results for surgical mask and fabric face mask not shown?

6. If the authors are confident that the test of M-74 in Fig. 1 is an outlier, please remove the data and provide justification in the discussion. Otherwise, it will be misleading if readers directly refer to the figure.

7. Page 5: “Participants were asked to don the mask…” –> “… do to mask…”?

8. Page 9, fit checks section, second paragraph: “Out of 35 tests on N95 masks, participants believed 17 masks fit, 2 with low confidence, 7 with medium confidence, and 7 with high confidence.” The total number does not match 17.

Reviewer #3: The authors studied the fit and filtration efficiency of masks and respirators. As written, the text contributes very little to the current understanding of respirators. The conclusion that, “Some respirators don’t fit” is not very helpful.

1. What is the reproducibility within one subject? That is, have them put it on and off.

2. Different races and ethnicities have different face shapes. Could this be impacting the fit?

3. How does fit change over time? Or with decontamination?

4. Qualitative fit testing can use odorants.

5. A positive control with a 3M 6200 respirator, i.e., painters respirator, might be helpful.

6. The figures are very blurry. Not publishable in current state.

6. PLOS authors have the option to publish the peer review history of their article (what does this mean?). If published, this will include your full peer review and any attached files.

Reviewer #1: No

Reviewer #2: No

Reviewer #3: No

---

## [Author Response · Author response to Decision Letter 0]

10 Nov 2020

We would like to thank the reviewers for their time and thoughtful suggestions. We have made changes addressing your comments and believe the changes have resulted in a more clear and impactful article. An list of changes made in response to your comments can be found in the letter to reviewers. For your convenience, we have written one letter per reviewer. Each letter addresses that reviewer's individual comments and concerns with the resulting changes and any other clarification or discussion.

---

## [Decision Letter · Decision Letter 1]

23 Nov 2020

PONE-D-20-25645R1

Comparing the Fit N95, KN95, Surgical, and Cloth Face Masks & Assessing the Accuracy of Fit Checking

PLOS ONE

Dear Dr. O'Kelly,

Thank you for submitting your manuscript to PLOS ONE. After careful consideration, we feel that it has merit but does not fully meet PLOS ONE’s publication criteria as it currently stands. Therefore, we invite you to submit a revised version of the manuscript that addresses the points raised during the review process.

We look forward to receiving your revised manuscript.

Kind regards,

Amitava Mukherjee, ME, Ph.D.

Academic Editor

PLOS ONE

Reviewers' comments:

Reviewer's Responses to Questions

**Comments to the Author**

1. If the authors have adequately addressed your comments raised in a previous round of review and you feel that this manuscript is now acceptable for publication, you may indicate that here to bypass the “Comments to the Author” section, enter your conflict of interest statement in the “Confidential to Editor” section, and submit your "Accept" recommendation.

Reviewer #4: All comments have been addressed

2. Is the manuscript technically sound, and do the data support the conclusions?

Reviewer #4: Yes

3. Has the statistical analysis been performed appropriately and rigorously? 

Reviewer #4: Yes

4. Have the authors made all data underlying the findings in their manuscript fully available?

Reviewer #4: Yes

5. Is the manuscript presented in an intelligible fashion and written in standard English?

Reviewer #4: Yes

6. Review Comments to the Author

Reviewer #4: General comments:

1. It is recommended to replace “N95 and KN95 masks” with “N95 and KN95 respirators” throughout the paper because masks are not designed for complete sealing.

Specific comments:

1. Page 4: the full name of CDC is the Centers for Disease Control and Prevention.

2. Page 12 and 17: it will be very helpful to illustrate the impact of minor facial differences on the fit of the respirators. It will be good to compare more fit factor scores on page 12, instead of qualitative comparison only. Also, please quantify the subcutaneous fat under the chin if possible. The authors may consider adding “bone structure and nose length and width” to page 17 as your examined factors.

7. PLOS authors have the option to publish the peer review history of their article (what does this mean?). If published, this will include your full peer review and any attached files.

Reviewer #4: No

---

## [Author Response · Author response to Decision Letter 1]

4 Jan 2021

Thank you for your commends and for continuing to help us revise the manuscript. We have made all minor changes requested, including changing "mask" to "respirator" when discussing N95 or KN95 masks, including "and Prevention" in the CDC's name, and adding detailed anatomical measurements to highlight how very small anatomical differences can result in significant qualitative fit factor differences.

---

## [Editor Report · Decision Letter 2]

6 Jan 2021

Comparing the Fit N95, KN95, Surgical, and Cloth Face Masks & Assessing the Accuracy of Fit Checking

PONE-D-20-25645R2

Dear Dr. O'Kelly,

We’re pleased to inform you that your manuscript has been judged scientifically suitable for publication and will be formally accepted for publication once it meets all outstanding technical requirements.

Kind regards,

Amitava Mukherjee, ME, Ph.D.

Academic Editor

PLOS ONE
---

## [Editor Report · Acceptance letter]

11 Jan 2021

PONE-D-20-25645R2 

Comparing the Fit of N95, KN95, Surgical, and Cloth Face Masks and Assessing the Accuracy of Fit Checking 

Dear Dr. O'Kelly:

I'm pleased to inform you that your manuscript has been deemed suitable for publication in PLOS ONE. Congratulations! Your manuscript is now with our production department. 

Kind regards, 

on behalf of

Professor Dr. Amitava Mukherjee 

Academic Editor

PLOS ONE